# Characterization of Hormone Receptor and HER2 Status in Breast Cancer Using Mass Spectrometry Imaging

**DOI:** 10.3390/ijms24032860

**Published:** 2023-02-02

**Authors:** Juliana Pereira Lopes Gonçalves, Christine Bollwein, Aurelia Noske, Anne Jacob, Paul Jank, Sibylle Loibl, Valentina Nekljudova, Peter A. Fasching, Thomas Karn, Frederik Marmé, Volkmar Müller, Christian Schem, Bruno Valentin Sinn, Elmar Stickeler, Marion van Mackelenbergh, Wolfgang D. Schmitt, Carsten Denkert, Wilko Weichert, Kristina Schwamborn

**Affiliations:** 1Institute of Pathology, School of Medicine, Technical University of Munich, Trogerstraße 18, 81675 Munich, Germany; 2German Cancer Consortium (DKTK), Partner Site Munich, 80336 Munich, Germany; 3Institute of Pathology, Philipps-University Marburg and University Hospital Marburg (UKGM), 35043 Marburg, Germany; 4German Breast Group (GBG), 63263 Neu-Isenburg, Germany; 5Department of Gynecology and Obstetrics, Comprehensive Cancer Center Erlangen-EMN, University Hospital Erlangen, Friedrich-Alexander University Erlangen-Nuremberg (FAU), 91054 Erlangen, Germany; 6Department of Gynecology and Obstetrics, Goethe-University Frankfurt, 60590 Frankfurt, Germany; 7Department of Obstetrics and Gynecology, University Hospital Mannheim, Medical Faculty Mannheim, Heidelberg University, 68167 Mannheim, Germany; 8Department of Gynecology, Universitätsklinikum Hamburg-Eppendorf, 20251 Hamburg, Germany; 9Mammazentrum Hamburg, 20357 Hamburg, Germany; 10Institute of Pathology, Charité-Universitätsmedizin, 10117 Berlin, Germany; 11Department of Obstetrics and Gynecology, University Hospital Aachen, 52074 Aachen, Germany; 12Klinik für Gynäkologie und Geburtshilfe, Universitätsklinikum Schleswig-Holstein, 24105 Kiel, Germany

**Keywords:** mass spectrometry imaging, breast cancer, proteomics, tissue typing, histopathology

## Abstract

Immunohistochemical evaluation of estrogen receptor, progesterone receptor, and human epidermal growth factor receptor-2 status stratify the different subtypes of breast cancer and define the treatment course. Triple-negative breast cancer (TNBC), which does not register receptor overexpression, is often associated with worse patient prognosis. Mass spectrometry imaging transcribes the molecular content of tissue specimens without requiring additional tags or preliminary analysis of the samples, being therefore an excellent methodology for an unbiased determination of tissue constituents, in particular tumor markers. In this study, the proteomic content of 1191 human breast cancer samples was characterized by mass spectrometry imaging and the epithelial regions were employed to train and test machine-learning models to characterize the individual receptor status and to classify TNBC. The classification models presented yielded high accuracies for estrogen and progesterone receptors and over 95% accuracy for classification of TNBC. Analysis of the molecular features revealed that vimentin overexpression is associated with TNBC, supported by immunohistochemistry validation, revealing a new potential target for diagnosis and treatment.

## 1. Introduction

Breast cancer, the most common cancer worldwide [1], is commonly characterized based on its (hormone) receptor status. This subtyping drives the diagnosis and subsequent choice of treatment. By evaluating the expression of estrogen receptor (ER), progesterone receptor (PR), and human epidermal growth factor (EGF) receptor-2 (HER2), pathologists can better predict the patient’s response to hormonal treatment or chemotherapy [2]. The majority of hormone receptor-positive breast cancer fall into the luminal A (ER+/PR+/HER2-/Ki67 low) and B (HER2-negative: ER+/PR-/HER2-/Ki-67 high; HER2-positive: ER+/PR any/HER2+/Ki67 any) subtypes [3,4].

Due to the lack of receptor response, triple-negative breast cancers (TNBC; PR-/ER-/HER2-) are often associated with worse clinical outcomes since they are not responsive to hormonal treatment. In those cases, patients are usually treated with neoadjuvant therapy; depending on the clinical tumor subtype, therapeutic backbones include endocrine therapy, immunotherapy, and chemotherapy [5]. Despite these efforts, patients frequently relapse within the first five years, leading to poor survival outcomes [6]. Tumor receptor status is routinely assessed by microscopic evaluation and quantification of immunohistochemistry (IHC) results, sometimes with additional in situ hybridization (ISH) assays, to clarify HER2 status [7]. These can be subject to variability due to different staining methods, antibodies used, and interpretation [8,9,10].

The limited treatment options for TNBC reflect the poorly known treatment targets for this subtype of breast cancer, which in turn explains the need to further comprehend the subclassification of TNBC and investigate new treatment opportunities [11].

Currently, tumor resection is one of the principal treatment modalities, with axillary lymph node (LN) metastasis being the most important prognostic factor for overall survival and tumor reoccurrence [12,13,14]. Therefore, accurate assessment of axillary LN is essential for breast cancer staging and choosing the appropriate therapeutic strategy [12,15]. Breast cancer diagnosis and prognosis have been improved by imaging methodologies that facilitate early detection of primary or metastatic lesions, discrimination of benign from malignant lesions, and guidance of intraoperative specimen evaluation [16]. Additionally, recent advances in omics technologies have led to an improved understanding of breast cancer pathobiology, molecular subtyping, and shed light on the tumor microenvironment and intra-tumoral heterogeneity [17,18,19].

Mass spectrometry imaging (MSI) has been applied in the field of oncologic pathology to map the distribution of analytes directly within intact tissue sections and to provide further insight into the nature and progress of tumor entities [20,21]. This methodology generates a mass spectrum for each measurement region within the tissue section, enabling the direct correlation with histological features without the requirement for external molecular targets such as antibodies. MSI studies on breast cancer, despite the low number of patient-derived samples, have yielded promising results [6,22,23,24,25,26]. Utilizing matrix-assisted laser desorption/ionization (MALDI) MSI combined with in situ tryptic digestion, we recorded proteomic data from 1191 patient samples which, in turn, were utilized to train and test machine-learning algorithms. Here, we present the results from classification models built for the characterization of the individual receptor status as a fast, inexpensive, and quantitative complementary approach to assist in the characterization of breast cancer. In addition, we have identified molecular features (vimentin and collagen) that, as result of their post-translational modifications, play an important role in the disease development, and which are therefore are potential treatment targets.

## 2. Results

### 2.1. Statistical Analysis and Classification

The obtained spectra data were subjected to a qualitative analysis of the peptide peaks concerning their signal-to-noise ratio and their distribution in the tissue prior to pre-processing. Based on the average spectra for the cohort, a feature list of the 376 most intense peptide features was created and employed for further analysis. The clinical and histological information regarding the epithelial regions were co-registered with the samples. Measurements and molecular pathology information regarding the receptor status were introduced in the attributes table for each patient. A separate attribute was added for all the cores that showed lower than normal activity for all three receptors, defined as TNBC. From an initial analysis of the average spectra of the different classes (Figure 1), some differences can be readily seen; with exception of HER2, where the differences are not recognizable solely by observation of the spectra.

To further evaluate the data, the proteomic signatures were randomly divided into test, validation, and classification subsets. The outcome of the classification on the validation set of different applied algorithms is summarized in Table 1. More detailed information about the classification results is provided in the Appendix A. The applied classification algorithms achieved a good performance when classifying the spectra, indicated by the high accuracy of the models.

Analysis of the receiver operating characteristic—area under the curve (ROC-AUC) highlights the features that best differentiate each class. Here, we show the five leading features (*m*/*z* values) per class (Table 2). Forward feature extraction, which selects the features per class based on their contribution to the classification was also determined and is shown in the Appendix A.

The distribution of the molecular features was analyzed to assure correlation with the histological annotations (Figure 2) before advancing with the fragmentation of the peptides for protein identification.

### 2.2. Protein Identification

The *m*/*z* values resulting from the ROC-AUC analysis were subjected to a literature search to identify the protein of origin. Features not present in the literature were also subjected to on-tissue fragmentation, which was carried out on whole-mount sections from the original patient blocks that had previously been used to build the tissue microarrays (TMAs).

Peptides with *m*/*z* 1198.7 corresponded to a fragment from actin (possible underlying isoforms: *ACTA1* or *ACTA2*, *ACTAB*, *ACTG1*, *ACTG2*, *POTEI*, *POTEKP*, *POTEF* or *POTEE)* [27].The *m*/*z* 772.4 molecular feature has been identified as a fragment of the Unc-79 homolog, NALCN channel complex subunit from direct fragmentation of the peptide in the tissue samples. This peak might also result from Collagen alpha-2(I) chain (*COL1A2*), as a cumulative effect with the second ionization peak at *m*/*z* 771.4. Furthermore, from direct on-tissue fragmentation of the peptides *m/z* 1428.7 and 1495.7, it was possible to conclude, after searching the MASCOT database, that both fragments are from vimentin. The results of the peptide identification are summarized in Table 3.

### 2.3. Immunohistochemical Validation

The overexpression of vimentin in TNBC samples was confirmed with IHC (Figure 3) utilizing an independent sample set consisting of two TMAs with TNBC (*n* = 51) and non-TNBC (*n* = 25) samples from the Institute of Pathology, Technical University of Munich, Germany. All non-TNBC samples showed no vimentin staining within the tumor cells, whereas 78% (*n* = 40) of all TNBC samples exhibited a positive vimentin staining within the tumor cells (17 (33%) strong positive; 15 (29%) medium positive; 8 (16%) weak positive) and 11 (22%) were negative. Of all TNBC cases, 36 were of no special type (NST), whereas the remaining TNBC cases were medullary (*n* = 8), lobular (*n* = 2), ductulolobular (*n* = 2), metaplastic (*n* = 1), apocrine (*n* = 1), and papillary (*n* = 1) (details can be found in Appendix A). Since androgen receptor positivity characterizes a molecular subtype of triple-negative breast cancer [30], we also performed staining for the androgen receptor. A total of 38 out of the 51 TNBC cases (75%) exhibited no androgen receptor positivity within the tumor cells, whereas 5 cases (10%) showed a strong androgen receptor positivity in almost all tumor cells, 6 cases an intermediate positivity in >1% of the tumor cells (12%), and 2 samples could not be evaluated due to missing cores. Of the 11 androgen receptor-positive cases, 7 did not exhibit any vimentin staining and 4 presented a medium or weak positive vimentin staining.

## 3. Discussion

Current methodologies to stratify the (hormone) receptor status in breast cancer samples rely on the evaluation of IHC staining as well as ISH in equivocal HER2 IHC results. IHC can quickly become very expensive when thorough molecular characterization is required to better define the tumor. Aside from being a significant financial burden, it is a qualitative and subjective approach, where evaluation depends on the laboratory procedure and pathologist’s expertise. These hindrances can be overcome by employing strategies that are able to perform unbiased relative quantification of tumor receptors. To develop a strategy that could aid with this predicament, we employed MSI, which enables the objective measurement of molecules directly in tissue sections. By training machine-learning algorithms with the collected data, we can assess the viability of the methodology to correctly evaluate (hormone) receptor status used for the clinical diagnosis and treatment stratification of breast cancer patients.

Our sample cohort consisted of samples collected from patients treated at different German healthcare institutions. However, to increase objectivity (by eliminating inter-observer bias), all samples used for statistical analysis were re-evaluated by two pathologists (A.N. and C.B.) who agreed on the attributed hormone receptor status. This step was necessary to generate more robust models because if we had utilized the assessment from different pathologists, we would have been perpetuating inter-observer bias and utilizing an imperfect sample set. This can, however, be recognized as a limiting factor of our approach, as the same pathologists cannot be continuously evaluating samples to expand the cohort.

The classification models were built using the average intensity of over 400,000 individual spectra to select the 376 most intense mass features from 1191 individual patient samples, which, to our knowledge, is the largest breast cancer cohort characterized by MSI to date. It should be further added that all samples were thoroughly annotated by two pathologists (C.B. and K.S.) so that solely the epithelial regions were considered for the classification. These were co-registered with the measurement regions and the respective relevant clinical information [31].

The classification models yielded accuracies of over 90% for ER and TNBC. However, for PR and HER2, the overall accuracy was lower in all classification models, indicating the lack of specific molecular features that can discriminate these receptors. The classification of TNBC had recognizably higher accuracy (>95%), being more accurate than the individual hormone receptor and HER2 status. This indicates that TNBC presents a more specific proteomic composition than the sum of its individual receptors. These findings suggest that further proteins/peptides, aside from the ones currently being employed for diagnosis, might have a central role in this malignancy and could therefore be used to better define tumor sub-types, and eventually be utilized for future therapeutic approaches. In our study, from the ROC-AUC analysis, it was possible to correlate the overexpression of vimentin (*m/z* 1428.7 and *m/z* 1495.79) with TNBC. This correlation has also been independently reported by works employing different analytical tools [18,19,20].

Despite the generally promising outcome from the ER, PR, and HER2 classification, the number of features extracted from the HER2-positive cases are rather scanty and mostly in the low *m/z* range. Additionally, these spectral features seem to contribute only marginally to the classification models, given their low accuracy. This contrasts with the results of a study by Rauser et al. wherein the HER2 status of breast cancer tissues could be determined by MSI utilizing fresh frozen tissue samples (*n* = 48) [22]. By employing support vector machine (SVM) and artificial neuronal network (ANN) as classification algorithms, 89% of the samples could be accurately classified as HER2-positive and HER2-negative, respectively. In particular, seven peptide ions showed clearly distinct intensities in the mass range of *m/z* 4740 to 8570. They were able to identify one of those *m/z*-values, namely *m/z* 8404, as cysteine-rich intestinal protein 1 (CRIP1). Due to the high molecular weight of the HER2 oncoprotein (185 kDa) and the covered mass range of 2.4–25 kDa, the authors did not succeed in the direct detection of HER2, but only indirectly via these protein profiles as surrogate markers. Since we employed in situ tryptic digestion to analyze FFPE (formalin-fixed paraffin embedded) tissue samples, the mass range of our analysis of tryptic peptides only extends to *m/z* 3200. Thus, we could not detect the same peptide ions proposed by Rauser et al. [22]. Since we performed trypsin digestion prior to measurement, the analysis rendered a higher number of peptides accessible to MALDI imaging in the lower mass range. A major limitation in this approach is, however, the fact that HER2 is a transmembrane protein. The amino acids lysine and arginine are found in a lower density in the proteins of the cell membrane [32]. As these amino acids represent the cleavage sites for trypsin, it is possible that HER2 was insufficiently digested and the only generated peptide fragments were too large to be captured by our measurements. In spite of all efforts to correctly classify HER2 on a proteomic level, one must consider that HER2-positive breast cancers can be highly heterogeneous [33]. HER2 is a receptor protein that displays a certain degree of plasticity, causing receptor activation and downstream signaling without gene amplification and protein overexpression [34]. These mechanisms of receptor activation include mutational alterations affecting the kinase or extracellular domain, and insufficiency of receptor dephosphorylation. Neither of these modifications can be detected by immunohistochemistry or in situ hybridization. Thus, these methods might not reveal the whole truth about the actual HER2 status, yet they are considered the gold standard and form the basis for critical assessment of new analytical methods such as mass spectrometry. Mass spectrometry, however, offers the possibility of identifying such protein modifications.

Other studies utilizing desorption electrospray ionization (DESI) MSI to analyze lipid profiles in order to determine hormone receptor as well as HER2 status in breast cancer tissue samples also could not achieve accurate classification results for HER2 status, while obtaining accurate results for ER and PR classification [35,36,37].

Moreover, as far as our cohort is concerned, features gained by the HER2-positive subgroup only had little influence on the classification algorithm. A similar pattern can be seen in the study by Brozkova et al. [38]. Hierarchical cluster analysis of invasive breast cancer of different types clearly outlined five subclasses. Several tumor characteristics such as nuclear grade, estrogen receptor alpha, progesterone receptor, triple-negative phenotype, Cyclin D1 immunohistochemistry, and Mucin 1 differed significantly between the created groups. HER2 status also differed between the groups but did not achieve statistical significance after Bonferroni correction for multiple testing. These findings imply that HER2 undoubtedly plays an important role in the diagnosis of breast cancer; yet, it is just one piece of the convoluted puzzle of diagnostic workup.

Further features could also be correlated with the different receptors. Overexpression of actin was correlated with ER and PR, as well as TNBC. The peptide corresponding to *m*/*z* 674.3 was overexpressed in ER- and PR-positive cases, while *m*/*z* 771.4 (Unc-79 homolog, NALCN channel complex subunit/Collagen alpha-2(I) chain) could also be associated with ER-positive cases and TNBC. Likewise, as specified earlier, vimentin overexpression was observed in epithelial cells of TNBC. Vimentin is highly expressed in epithelial tumors (prostate cancer, gastric cancer, malignant melanoma, and lung cancer); however, in breast cancer, aberrant expression of vimentin is restricted to TNBC [39]. Vimentin, an intermediate filament protein, regulates the epithelial mesenchymal transition, and highly impacts the cellular proliferation and the invasiveness of the disease [40,41]. These findings were corroborated by the IHC analysis, performed in an external cohort collected at the Technical University of Munich, which identified vimentin in the epithelial tumor region of TNBC, while in the luminal and HER2+ subtypes, vimentin was solely present in the stroma. Despite the high levels of vimentin in the tumor cells of TNBC, a commentary should be added with regard to its consistency. In the validation dataset, we observed that the expression of vimentin is not always strongly detected by IHC, which can be related to tumor biology. With respect to these findings, a systematic evaluation of the expression of this protein in TNBC should be considered to grant a more robust interpretation of its activity and potential diagnostic and prognostic capabilities. Similarly, additional investigation about the role of collagen in different subtypes of breast cancer should be carried out by employing specific proteomic digestion utilizing collagenase, which could better delineate the specificities between different breast cancer subtypes.

Molecular classification of breast cancer has been refined more and more in recent years to reliably predict the outcome and response to therapy. Nevertheless, there remains a varying degree of heterogeneity within molecular subtypes. This is particularly true for TNBC, and is underscored by the fact that there are several established subgroups for this breast cancer subtype alone. As we can see on the classifications, TNBC was more accurately (95.9% LDA, 97.9% RF, and 98.7% KNN) identified than the individual receptors independently. These findings, in such a large cohort, indicate a strong possibility that other peptides/proteins can be targeted for the identification and evaluation of TNBC subtypes, similar to molecular subtypes in TNBC identified by genomic sequencing [42]. In addition, a more comprehensive molecular classification of hormone receptor-negative tumors could expose new treatment targets and, consequently, new tumor subtypes. In a proof of concept study by Holzlechner et al. [43], MALDI MSI was successfully applied to colon tissue to detect and examine tissue-resident immune cells. On the basis of a set of *m/z* features, the distribution of T-lymphocytes within nodular lymphoid aggregates and macrophages as dispersed single cells within the *lamina propria* was vividly displayed and correlated well with the immunohistochemical stains for CD3 and CD206. Thus, it seems obvious that MALDI MSI can serve as a promising tool to also investigate tumor infiltrating lymphocytes (TILs) that largely contain lymphocytes and macrophages. Due to the size of this cohort and, thus, its large range of morphological immune response patterns, we intend to also focus on the characterization of TILs in the next steps. Similarly, additional investigation about the role of collagen in the different subtypes of breast cancer should be carried out by employing collagenase in order to better delineate the specificities between different breast cancer subtypes.

## 4. Materials and Methods

### 4.1. Patients and Data Collection

The 1191 patient samples measured were part of the German Adjuvant Intergroup Node-Positive (GAIN) 1 study. The GAIN-1 study (ClinicalTrials.gov NCT00196872) was a prospective multicenter phase III trial to compare two dose-dense (dd) regimens: intensified dd epirubicin, paclitaxel, and cyclophosphamide (EPC) versus dd epirubicin, cyclophosphamide, paclitaxel, capecitabine (EC-PwX), and ibandronate versus observation in patients with high-risk, node-positive primary breast cancer (BC). In addition, radiotherapy, endocrine treatment, and adjuvant trastuzumab (starting 05/2006) were given according to recommendations of the national ‘*Arbeitsgemeinschaft für Gynäkologische Onkologie* (Association for Gynecological Oncology, AGO)’ guideline. Patients with histologically confirmed, unilateral, or bilateral primary node-positive BC were enrolled after providing written informed consent for clinical trial participation and use of biomaterials. Patients needed to have received adequate surgical treatment with histological complete resection (R0) of the primary tumor and 10 resected axillary nodes as per the standard of care at the time of conducting the study. Overall, the trial recruited 3023 patients between 2004 and 2008, and 2994 patients were assigned for initial treatment. Clinicopathological data were extracted from the clinical study database. Ethical committee approval from all centers participating in the clinical study and from the Institutional Review Board of Charité University Hospital Berlin (Germany) was obtained. Institutional Ethical committee approval from the University Hospital Marburg was obtained (approval 121/20 from 5 October 2020). This study was conducted adhering to the REMARK (Reporting Recommendations for Tumor Marker Prognostic Studies) criteria [44]. GAIN-1 and study cohort local pathology reports are summarized in Appendix A.

The samples herein analyzed were collected at different clinical institutions across Germany and assembled in tissue microarrays (TMAs). One core from the resected archival sample of each patient was added to the TMAs. The patient samples were distributed across a total of 13 TMAs.

The IHC validation cohort is comprised of duplicate samples from 51 patients with TNBC and 25 patients with non-TNBC (16 luminal, 9 HER2+) collected between 2000–2020, randomly distributed across two different TMAs.

### 4.2. Assessment of Hormone Receptor and HER2 Status

The evaluation of the hormone receptors (ER, PR) was performed semi-quantitatively, using the WHO (World Health Organization)-recommended agonists for the respective receptors according to the following scheme, where the percentage represents the proportion of positive tumor cells in the tissue sample:0 = 0%, 1 = 1%, 2 = 2–5%, 3 = 6–10%, 4 = 11–50%, 5 = 51–80%, 6 > 81%

Scoring of HER2 was performed according to the American Society of Clinical Oncology/College of American Pathologists guideline recommendations for HER2 (Wolff et al., 2018). The evaluation was primarily based on immunohistochemistry followed by in-situ hybridization (ISH) in equivocal cases. Positive HER2 cases are defined as those with >10% of tumor cells showing homogeneous, dark circumferential (chicken wire) staining patterns (score 3+). Negative HER2 cases are regarded as those with no staining (score 0) or membranous staining that is incomplete, faint/barely perceptible, and in cases ≤ 10% of tumor cells (score 1+). Weak to moderate complete membrane staining in > 10% of tumor cells is regarded as an ambiguous result (score 2+) that needs further clarification by in situ hybridization techniques; in our case, dual-probe chromogenic in situ hybridization. The evaluation is performed according to the following criteria: HER2/CEP17 ratio ≥ 2 and average HER2 copy number ≥ ≥ 4.0 signals/cell: ISH positive; HER2/CEP17 ratio ≥ 2 and average HER2 copy number < < 4.0 signals/cell: ISH negative; HER2/CEP17 ratio < 2 and average HER2 copy number ≥ ≥ 6.0 signals/cell: ISH positive; HER2/CEP17 ratio < 2 and average HER2 copy number ≥ ≥ 4.0 and < < 6.0 signals/cell: ISH negative; HER2/CEP17 ratio < < 2 and average HER2 copy number < < 4.0 signals/cell: ISH negative.

### 4.3. Proteomic Characterization by Matrix Assisted Laser Desorption/Ionization Mass Spectrometry Imaging

For this study the cohort consisted of formalin-fixed paraffin embedded (FFPE) specimens from patients (*n* = 1191), of which 106 samples were excluded from the statistical analysis due to low tumor content or poor tissue fixation. The samples were collected from blocks of patients diagnosed as luminal A (*n* = 596) luminal B (*n* = 121), HER2+ (*n* = 200), and TNBC (*n* = 105). A total of 6% of the samples had an inconclusive outcome for at least one receptor status but not for others; thus, these were included in the analysis of the individual receptor (hormone receptors and HER2 internal pathology assessments are summarized in Appendix A). Diagnosis was made according to the 4th Edition WHO classification of breast cancer (2012), and the local diagnosis information has been summarized in Appendix A. More recent WHO guidelines are currently in place for the classification of tumors of the breast [45]. Nonetheless, for the presented study, solely the evaluation of ER, PR, and HER2, as described in the previous section, has been used to group the samples, and no investigations with regard to the histological subtype of the tumor were performed. The samples were assembled in tissue microarrays (TMAs), from which a slice of 4 μm was adhered to an indium-tin oxide (ITO) slide (Bruker Daltonics, Bremen, Germany), Figure 4(1). Sample preparation has been previously reported [46,47]. Briefly, as depicted in Figure 4, after heating the slides at 95 °C for 10 min, samples were dewaxed with xylene (Carl Roth GmbH, Karlsruhe, Germany), rehydrated through graded ethanol washes (Carl Roth GmbH), and subjected to heat-induced antigen retrieval in MilliQ water at 95 °C for 20 min. For on-tissue digestion, a trypsin (Promega, Mannheim, Germany) solution was prepared to a final concentration of 0.1 µg/µL. The enzyme solution was sprayed on with an automatic sprayer (TM sprayer, HTX Technologies, Chapel Hill, NC, USA) in 25 cycles with a fixed spraying flow of 150 μL/min. Sections were subsequently incubated in a humidity chamber at 37 °C for 2 h. Following digestion, the samples were again laid in the spraying chamber and sprayed with 10 mg/mL of α-cyano-4-hydroxycinnamic acid matrix (Sigma-Aldrich, Chemie GmbH, Munich, Germany) in 70% acetonitrile aqueous solution with 1% trifluoracetic acid (Carl Roth GmbH) at a set flow of 120 μL/min.

MSI was performed using a Rapiflex™ MALDI-time-of-flight (TOF) mass spectrometer (Bruker Daltonics). A peptide calibration standard mix including bradykinin, angiotensin II, angiotensin I, substance P, bombesin, ACTH clip 1–17, ACTH clip 18–39, and somatostatin 28 (Bruker Daltonics) was used for external calibration. Each spectrum was automatically generated at a spatial resolution of 50 µm using flexControl (Bruker Daltonics) in the mass range of *m/z* 600–3200. A total of 1000 laser shots were acquired for each spectrum at 1 kHz laser frequency. The measurement regions were defined using flexImaging (Bruker Daltonics). Following MSI measurements, the matrix was removed by two washes in 99.99% methanol (Carl Roth GmbH) for 3 min each, followed by two washings in 99.99% ethanol (Carl Roth GmbH) for 10 s.

### 4.4. Tumor Annotation, Data Processing, and Extraction

Tissue sections analyzed by MSI were stained with H&E (hematoxylin and eosin) and digitalized with a slide scanner (Aperio CS2, Leica Biosystems, Wetzlar, Germany). The H&E scans were uploaded and the epithelial tumor regions were thoroughly annotated (B.C. and K.S.) using SCiLS Cloud (discontinued service from Bruker Daltonics; a recommended alternative is the use of QuPath and its integration with SCiLS Lab, from Bruker Daltonics). MSI data was processed using SCiLS Lab MVS (Version 2023a Premium 3D, Bruker Daltonics) for mass spectrometry and image visualization. Annotations were imported into SCiLS Lab MVS software (Version 2023a Premium 3D). Spectra baseline was corrected to the total ion count (TIC). Afterward, spectra were pre-processed for intensity profile normalization, re-sampling, spatial de-noising, and calculation of a second normalization profile [48,49]. Subsequently, automated peak-picking (376 most intense peaks) in the range of 600–3000 Da range was performed, and the spectra for the individual spots were exported to .csv- format and imported to R statistical software (version 3.6.3) and RStudio 1.2.5033 [50].

### 4.5. MS/MS Measurements

Tentative identification of the *m*/*z* features was carried out by MS/MS measurement on whole-mount breast cancer tissue sections. The measurements were performed in situ using a Rapiflex in LIFT mode (*m*/*z* 772.4 and 1198.7) and a timsTOFflex (*m*/*z* 1428.7 and 1495.7) mass spectrometer (Bruker Daltonics). Laser power for fragmentation was set at 70–80% in positive ionization mode, 2000 shots at a laser frequency of 5 kHz, and a beam scan of 25 µm^2^. For the identification, the MS/MS spectra (Appendix A) were submitted to MASCOT MS/MS Ion Search, where the SwissProt database was searched to match tryptic peptide sequences to the respective intact proteins, defining *homo sapiens* as the taxonomic class. The MS/MS spectrum search parameters included a mass tolerance of 1 Da, MS/MS tolerance of ±1 Da, up to two missed cleavages, methionine oxidation, protein *N*-terminal acetylation, and proline oxidation as variable modifications.

### 4.6. Statistical Analyses

#### Supervised Classifications

Linear Discriminant Analysis (LDA) classification was performed using the “caret” package for R. The dataset was split into training (80%) and test (20%) sets. The data was fitted using the method “lda”. The method control was set to 5-fold cross-validation. Random Forest (RF) classifiers were built using the “caret” package for R. The dataset was split into training (80%) and test (20%) sets. The data was fitted using the method “ranger”, with 25 trees. The method control was set to 5-fold cross-validation. K-nearest neighbors (KNN) classifiers were built using the “caret” package for R. The dataset was split into training (80%) and test (20%) sets. The data was fitted using the method “knn”. The method control was set to 5-fold cross-validation.

Similarly, for the Support Vector Machine (SVM) classification, the dataset was split into training (80%) and test (20%) sets. The data was fitted using the method “svmRadial”. The method control was set to 5-fold cross-validation. The tune length was set to 5 values of automatic grid search. Using the “caret” package, the fitted model was tested on test subset and the accuracy was calculated based on the results of the confusion matrix. SVM was tuned to the best c factor (c = 16 for ER and PR and c = 32 for HER2 and TNBC).

All fitted models were utilized to classify the test subset data. The accuracy of each model was calculated based on the results of the confusion matrix.

### 4.7. Immunohistochemical Analysis

For validation, immunohistochemical staining was performed utilizing a different patient cohort. Two TMAs containing human breast cancer samples were constructed from archived breast 1lonal mouse antibody, clone V9, dilution 1:300, Agilent Dako, Agilent Technologies, Santa Clara, CA, USA) and androgen receptor (monoclonal rabbit antibody, clone SP107, dilution 1:10, Cell Marque, Merck KGaA, Darmstadt, Germany) staining was carried out on a Ventana Benchmark XT autostainer (Roche Diagnostics, Rotkreuz, Switzerland). All stained slides (vimentin, androgen receptor, and H&E) were digitalized utilizing a slide scanner (Aperio AT2 slides scanner, Leica Biosystems). Staining results were evaluated by a board-certified pathologist (K.S.) and qualitatively classified in a four-tiered grading system for vimentin (negative, weak positive, medium positive, and strong positive) to enable comparability (Appendix A).

## 5. Conclusions

Breast cancer continues to be one of the most diagnosed types of cancer among women worldwide. Receptor-negative breast tumors lack treatment options due to unknown viable treatment targets. In this study, we showed that MSI is an advantageous technique to support sub-typing of breast cancer and an important tool for the discovery of novel tissue markers. Vimentin could be identified as a protein overexpressed in tumor cells of triple-negative breast cancer, unlike their receptor-positive counterparts. We foresee that further investigations into the role of vimentin will disclose new pathways of this disease development and novel treatment targets.

## Figures and Tables

**Figure 1 ijms-24-02860-f001:**
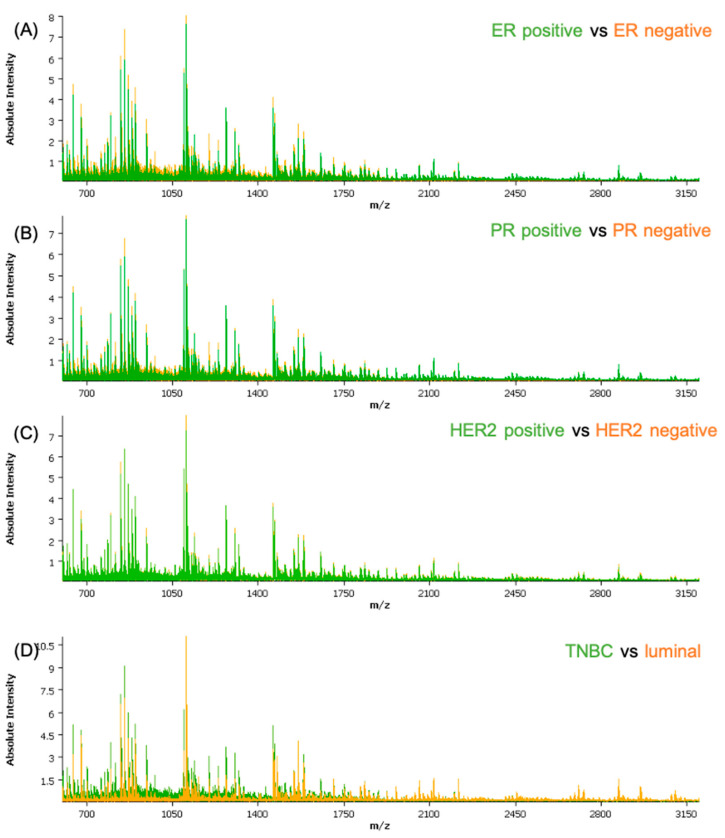
Comparison between the average spectra of the different classes used in the classification models: (**A**) ER-positive versus ER-negative; (**B**) PR-positive versus PR-negative; (**C**) HER2-positive versus HER2-negative; and (**D**) TNBC versus luminal. ER—estrogen receptor; PR—progesterone receptor; HER2—human epidermal growth factor receptor-2.

**Figure 2 ijms-24-02860-f002:**
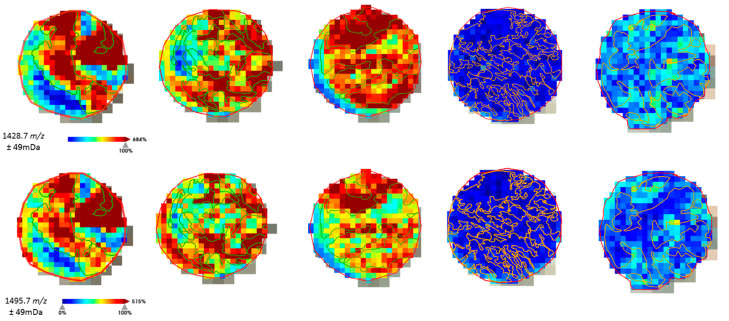
Example of the on-tissue distribution of a few molecular features obtained by ROC-AUC calculation. Histological annotation of the tumor regions (TNBC in green, non-TNBC in orange) could be correlated with the distribution intensity of the peptide fragments selected for TNBC.

**Figure 3 ijms-24-02860-f003:**
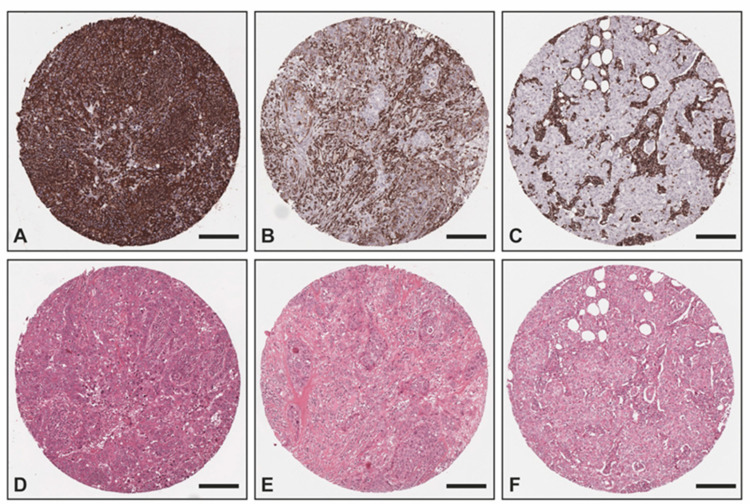
Exemplary images of vimentin IHC and corresponding H&E: (**A**,**D**) TNBC with strong positivity within the tumor cells as well as the tumor stroma; (**B**,**E**) TNBC with intermediate positivity within the tumor cells and strong positivity within the tumor stroma; (**C**,**F**) non-TNBC with no vimentin staining within the tumor cells and strong positivity within the tumor stroma. Scale bar—200 μm.

**Figure 4 ijms-24-02860-f004:**
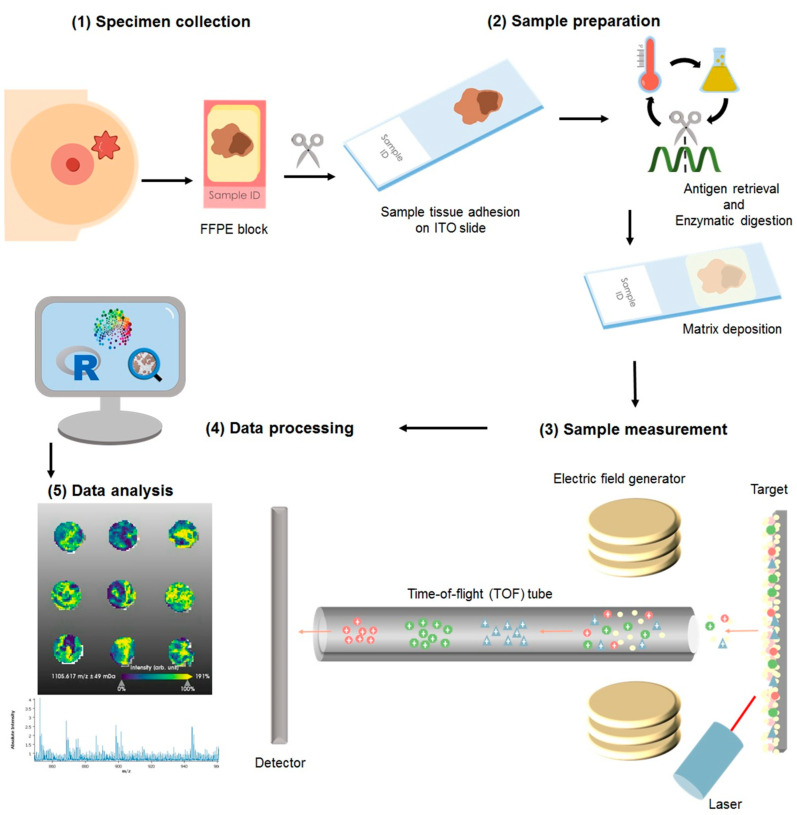
Sample processing for matrix-assisted laser desorption/ionization mass spectrometry imaging: (**1**) The resected specimen is fixed in formalin and embedded into paraffin as per standard sample treatment for histopathological evaluation and clinical diagnosis. From the blocks, a section is cut and adhered to an indium-tin-oxide (ITO) slide. (**2**) The sample is then subjected to enzymatic digestion, followed by matrix application with a solution sprayer. (**3**) The samples are measured using a MALDI-TOF system. (**4**) The acquired data is subjected to pre-processing where the spectra are normalized and re-sampled. (**5**) The sample is stained with hematoxylin and eosin to facilitate histological annotation of the regions of interest, followed by co-registration with the mass spectrometry measurements. The data is then subjected to statistical analyses.

**Table 1 ijms-24-02860-t001:** Classification outcome. LDA—Linear discriminant analysis; RF—Random Forest; KNN—k-nearest neighbors; SVM—Support vector machine.

Classification Results (%)				
**Model**	**ER**	**PR**	**HER2**	**TNBC**
**LDA**	90.6	81.2	77.7	95.9
**RF**	95.5	92.2	93.4	97.9
**KNN**	97.3	95.2	95.2	98.7
**SVM**	95.56	90.81	88.81	98.79

**Table 2 ijms-24-02860-t002:** Receiver operating characteristic—area under the curve (ROC-AUC) feature comparison.

ER		PR		HER2		TNBC	
*m/z*	Accuracy	*m/z*	Accuracy	*m/z*	Accuracy	*m/z*	Accuracy
1198.73	0.816	1198.73	0.704	606.12	0.616	1198.73	0.849
1199.71	0.780	1199.71	0.691	668.00	0.602	1428.74	0.812
772.39	0.775	674.35	0.673	630.08	0.601	772.39	0.810
674.35	0.768	772.39	0.671	645.07	0.598	1199.71	0.806
805.43	0.766	1200.68	0.669	628.08	0.595	1495.79	0.800

**Table 3 ijms-24-02860-t003:** Summary of the tentative peptide identification by in situ MS/MS analysis.

*m/z*	Mr (Expt)	Tentative ID	Sequence	MASCOT Score
1198.7	1198.71	Actin *	AVFPSIVGRPR	47
805.4	805.46	Collagen alpha-3 (VI) chain	ALEFVAR	^#^
771.4	771.42	*UNC79*	GPVESKR	40
771.4	771.41	Collagen alpha-2 (I) chain	GASGPAGVR	
1428.7	1427.7083	Vimentin	SLYASSPGGVYATR	81
1495.7		Vimentin	TYSLGSALRPSTSR	41

* Possible underlying isoforms: ACTA1 or ACTA2, ACTAB, ACTG1, ACTAG2, POTEI, POTEKP, POTEF or POTEE. ^#^ Tentative identification based on literature search. ^#^ from literature search [28,29].

## Data Availability

The data presented in this study are available on request from the corresponding author.

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
