# Peer review of "Characterization of Hormone Receptor and HER2 Status in Breast Cancer Using Mass Spectrometry Imaging"

_ijms, 2023, doi:10.3390/ijms24032860_

Round 1

Reviewer 1 Report

Juliana and colleagues describe a mass spectrometry imaging method to detect ER,HR and HER2 in breast cancer. Despite multiple well-established mass spectrometry, the advances generated from this paper is minor. Most of the data reported in this paper add therefore minimal information to previous knowledge, and do not bring new scientific concept. Furthermore, the writing style, organization and annotation of the data makes it hard to follow.

Author Response

The comment made by the reviewer shows a limited knowledge about the current challenges in the histopathologic evaluation of tumors, namely breast cancer, and the utilization of technologies such as mass spectrometry imaging (MSI) in cancer research. We not only show that MSI is a promising technique for the analysis of tumor tissue, as we have trained algorithms for the classification of the breast tissue using the largest cohort, to our knowledge, of samples collected across different German healthcare institutions for the evaluation of this tumor type. We could also show that this is possible by utilizing just one single tissue section that can afterward still be utilized for further analysis, for instance, stained with H&E for histopathology evaluation (in our case, but other types of staining including immunohistochemistry are also possible) or additional investigation with other techniques. Furthermore, we show that we can extract meaningful features, in a spatially resolved manner, that can aid in tumor diagnosis, prognosis, and subsequently can identify possible targets of interest for novel therapies. In this case, we show that vimentin has a central role in the management of triple-negative breast cancer. This result was also validated by IHC on a second independent sample set.

When this scientific advances in clinical research are deemed “minimal”, we must consider this reviewer inapt to place judgment of the presented work, or that some conflict of interest, which the authors are unaware of, prevents the reviewer from being objective and unbiased.

As for the writing and English language, the paper was written and edited by several authors well-versed in English grammar. Nonetheless, we have further revised the spelling and grammar. 

Reviewer 2 Report

Dear Authors:

The manuscript "Characterization of Hormone Receptor and HER2 status in Breast Cancer using Mass Spectrometry Imaging" by Gonçalves et al has demonstrated that MSI is an advantageous technique to support sub-typing of breast cancer and an important tool for the discovery of novel tissue markers. Vimentin could be identified as a protein overexpressed in tumor cells of triple negative breast cancer, unlike their receptor positive counterparts. I have just a few suggestions.

Some references or background information is missing. In introduction, please add more background information aboutn breast cancer and axillary lymph node metastasis. Some articles has also demonstrated it. (Please cite: 1. Robot-Assisted Minimally Invasive Breast Surgery: Recent Evidence with Comparative Clinical Outcomes. J Clin Med. 2022 Mar 25;11(7):1827. doi: 10.3390/jcm11071827.

2. Efficacy of da Vinci robot-assisted lymph node surgery than conventional axillary lymph node dissection in breast cancer - A comparative study. Int J Med Robot. 2021 Dec;17(6):e2307. doi: 10.1002/rcs.2307. Epub 2021 Jul 29. PMID: 34270843.

3. Patient Management Strategies in Perioperative, Intraoperative, and Postoperative Period in Breast Reconstruction With DIEP-Flap: Clinical Recommendations. Front Surg. 2022 Feb 15;9:729181. doi: 10.3389/fsurg.2022.729181. PMID: 35242802; PMCID: PMC8887567.)

Best,

Author Response

We appreciate the review’s observation which found a gap in our introduction, which we have now adapted with the inclusion of the suggested bibliography.

Reviewer 3 Report

The work of Juliana Pereira Lopes Gonçalves and collaborators aims to use mass spectrometry imaging in breast carcinomas in order to stratify these carcinomas into the four common biological classes: luminal A and B, HER2 enriched, and TNBC. Furthermore, the authors examine additional immunohistochemical markers in order to characterize the TNBC subpopulation by identifying vimentin as a potential target.

Although the argument is in line with the journal's objectives and has good scientific validity, in my opinion this manuscript has some gaps that need to be filled.

1) Line 344: the sum of the samples declared usable is 1022. If the 136 excluded are added to these, the total sum is equal to 1159 and not 1191.

Furthermore, it is not clear whether the histological material used comes from tissue biopsies or from total resection of the tumor, just as it is not clear whether for some patients multiple tumor inclusions originating from the same tumor site or from multiple tumor sites were analysed.

It is also not clear why the histological analysis of the cases was done according to the 2014 WHO guidelines and not according to the more recent 2019 ones.

2) There is no table that allows evaluating the histotype, Grade, TNM and receptor status of the cases examined. This is also essential to better evaluate the subclass of TNBC carcinomas within which there may be histotypes which by definition do not express receptor markers (e.g. metaplastic carcinoma) as well as histotypes, which usually can also express them.

3) As also stated by the authors in the discussion section, the marker for vimentin is expressed in those carcinomas that are usually poorly differentiated and tend to acquire a mesenchymal transition. However, it is not clear why vimentin expression correlates with the triple negative phenotype while vimentin positive TNBC tumor variants are present in the literature. Did the authors try to correlate the presence of Vimentin with regards to the different tumor histotypes that make up the analyzed TNBC population? was the androgen receptor marker also tested to identify apocrine histotype? In fact, the latter can be positive for vimentin.

Minor revision:

-Line 281: double check the markers

- Line 59: please cite Grassini et al (DOI: 10.1159/000524227)

- Line 99: please cite Annaratone et al (DOI: 10.1159/000507055)

Author Response

The authors would like to acknowledge the reviewer for the time spent reviewing our work. We have carefully considered your comments, which we have addressed.

1) We appreciate the keen observation. The values are not adding up because, as we describe in the very same methodology section, some of the samples (6% of the patient cohort) could not be grouped in the four subgroups mentioned since one or more than one receptor status was inconclusive, but could still be included to train the models of the defined receptor evaluation.

All samples were obtained from resection specimens of patients previously diagnosed with breast cancer. Surgery was performed at different clinical institutions all across Germany. Samples were grouped in TMAs containing one core per patient. In light of the reviewer’s comment, we considered that the paragraph within the manuscript might not be as clear as we initially thought, and, therefore, added another sentence further highlighting the cohort composition. The patient cohort is composed of archival samples collected prior to 2019, therefore the guidelines applied were the WHO 2014.

2) We thank the reviewer for her/his suggestion to include a table with information regarding tumor type, grade, receptor status, and TNM. The local pathology reports were compiled and have been summarized in table S3, as well as further information concerning the cohort has been added to the supplementary.

3) This is a very interesting suggestion, which we do intend to investigate further. This initial study did indeed raise several questions regarding the role of vimentin in the different compartments of the tissue, namely in the epithelial regions of TNBC. We intend to follow up and do a systematic investigation to further understand and perhaps try to establish a solid correlation between the clinical data and the vimentin expression. We, however, understand that it is (and will be) a complex analysis and at the moment our answers in this regard are still limited. Nonetheless, the reviewer’s suggestion will also be considered, and we hope to better address this in a future publication.

As for this manuscript, we have added the information in histotypes/subtypes for all TNBC cases present in the validation set. The detailed information can be now found in supplemental table 4. Also, we performed additional staining for androgen receptor. In total, 11 cases showed androgen receptor positivity (including the one apocrine carcinoma), but only 4 of these also exhibited a medium or weak positive vimentin staining. This information has been added to the manuscript.

The minor comments have also been addressed and we thank the reviewer for enriching our citation list. 

Round 2

Reviewer 3 Report

The authors have followed most of my suggestions and many improvements have been made. However there are still some substantial changes that need to be made. It is the opinion of this reviewer that the choice to use a 2014 WHO classification to histologically identify previous cases is unjustified. Readers are interested in acquiring new scientific information based on the most up-to-date criteria.

Compared to 2014, in the most updated 2019 guidelines, many histotypes have been re-evaluated and above all the method of identifying mixed tumors has changed.

Scientific assessments of the manuscript should be re-evaluated following a 2019 WHO reclassification of carcinomas by a pathologist.

Author Response

The authors would like to thank the reviewer for the input, which has been carefully considered and discussed among the authors. While the existence of more recent guidelines is not lost on the highly knowledgeable panel of pathologists authoring this manuscript, there are no claims being made considering the histological subtypes of the tumor. In fact, solely the tumor biology, and its regards to the individual evaluation of the hormone receptors and HER2 has been considered for the presented investigation. Thus, reevaluation of the histological subtypes would not change the outcome of the study, nor allow for a better understanding of the approach. We do however consider that the reviewer has a point in that the reader should be provided with the most up-to-date information, and, in that regard, we have added an explanation to the manuscript that more recent guidelines are available, with the appropriate reference, thus, providing the most up-to-date criteria.

Round 3

Reviewer 3 Report

The explanations of the authors were exhaustive. Publication in this version is recommended.